# Nanomedicine in the Management of Alzheimer’s Disease: State-of-the-Art

**DOI:** 10.3390/biomedicines11061752

**Published:** 2023-06-18

**Authors:** Shehla Nasar Mir Najib Ullah, Obaid Afzal, Abdulmalik Saleh Alfawaz Altamimi, Hissana Ather, Shaheen Sultana, Waleed H. Almalki, Pragya Bharti, Ankit Sahoo, Khusbu Dwivedi, Gyas Khan, Shahnaz Sultana, Abdulaziz Alzahrani, Mahfoozur Rahman

**Affiliations:** 1Department of Pharmacognosy, Faculty of Pharmacy, King Khalid University, Abha 62529, Saudi Arabia; shehlanasar2005@gmail.com; 2Department of Pharmaceutical Chemistry, College of Pharmacy, Prince Sattam Bin Abdulaziz University, Alkharj 11942, Saudi Arabia; obaid263@gmail.com (O.A.); alfawazaltamimi123@gmail.com (A.S.A.A.); 3Department of Pharmaceutical Chemistry, King Khalid University, Abha 62529, Saudi Arabia; hissana@kku.edu.sa; 4IIMT College of Pharmacy, Greater Noida 201310, Uttar Pradesh, India; shaheen634@yahoo.co.uk; 5Department of Pharmacology and Toxicology, College of Pharmacy, Umm Al-Qura University, Makkah 21955, Saudi Arabia; whmalki@uqu.edu.sa; 6MM College of Pharmacy, Maharishi Markandeshwar (Deemed to Be University), Mullana 133207, Haryana, India; bhartipragya62@gmail.com; 7Department of Pharmaceutics, Pharmaceutical Sciences, Shalom Institute of Health & Allied Sciences, Sam Higginbottom University of Agriculture, Technology & Sciences, Allahabad 211007, Uttar Pradesh, India; ankitsahoo71@gmail.com; 8Department of Pharmaceutics, Sambhunath Institute of Pharmacy Jhalwa, Prayagraj 211015, Uttar Pradesh, India; khusbudwivedi99@gmail.com; 9Department of Pharmacology, College of Pharmacy, Jazan University, Jazan 45142, Saudi Arabia; gkhan@jazanu.edu.sa; 10Department of Pharmacognosy, College of Pharmacy, Jazan University, Jazan 45142, Saudi Arabia; shmali@jazanu.edu.sa; 11Pharmaceuticals Chemistry Department, Faculty of Clinical Pharmacy, Al-Baha University, Alaqiq 65779-7738, Saudi Arabia; alzahraniaar@bu.edu.sa; 12Department of Pharmaceutical Sciences, Shalom Institute of Health & Allied Sciences, Sam Higginbottom University of Agriculture, Technology & Sciences, Allahabad 211007, Uttar Pradesh, India

**Keywords:** Alzheimer’s disease, synthetic drugs, adverse effects, nanomedicines, challenges, clinical development

## Abstract

Alzheimer’s disease (AD) is a deadly, progressive, and irreversible brain condition that impairs cognitive abilities. Globally, it affects 32.6 million individuals, and if no viable therapies are available by 2050, that figure might rise to 139 million. The current course of treatment enhances cognitive abilities and temporarily relieves symptoms, but it does not halt or slow the disease’s development. Additionally, treatments are primarily offered in conventional oral dosage forms, and conventional oral treatments lack brain specialization and cause adverse effects, resulting in poor patient compliance. A potential nanotechnology-based strategy can improve the bioavailability and specificity of the drug targeting in the brain. Furthermore, this review extensively summarizes the applications of nanomedicines for the effective delivery of drugs used in the management of AD. In addition, the clinical progress of nanomedicines in AD is also discussed, and the challenges facing the clinical development of nanomedicines are addressed in this article.

## 1. Introduction

Alzheimer’s disease (AD) is a neurological condition that progressively affects neurons and causes irreversible dementia syndrome [1,2]. The incidence of AD grows with age and is more common in older people [3]. It currently affects 32.6 million people worldwide, but without new treatments, it is expected to nearly double every 20 years, reaching 78 million by 2030 and 139 million by 2050 [4]. It causes a decline in behavioral, mental, and other intellectual abilities. It leads to the death of brain cells, which causes memory loss [5]. AD is a neuropathological disease caused by the extracellular aggregation of amyloid beta (Aβ) plaques and the accumulation of intraneuronal neurofibrillary tangles of tau (τ) protein [6]. Aβ is developed by the division of type I membrane proteins, such as amyloid precursor protein (APP), and is present at significant levels in the patient’s cerebrospinal fluid and plasma. Therefore, depletion of Aβ plaque deposition and inhibition of p-τ fibrils is involved in the dual-target therapy for AD. It is an uncurable disorder and does not have specific diagnostic approaches or therapeutic drugs. The therapeutic agents presently known for AD only reduce the symptoms [7]. The current course of treatment enhances cognitive abilities and temporarily relieves symptoms, but it does not halt or slow the disease’s development. Additionally, treatments are primarily offered in conventional oral dose forms, and conventional oral treatments lack brain specialization and cause adverse effects, resulting in low patient compliance.

AD is also a degenerative metabolic disease associated with cardiovascular risk factors such as hypertension, diabetes, atherogenic dyslipidemia, and obesity [8]. Environmental exposure to toxicants, genetic factors, mutation, trauma, and metabolic diseases, such as diabetes mellitus and obesity, can cause AD [9]. The new potential of nanotechnology is leading to new possibilities for diagnosing, treating, and curing human diseases [9,10]. Nanotechnology is employed for drug delivery development to treat the most complex diseases, such as cancer and neurodegenerative disorders. Nanotechnology can help in the early detection of AD via highly efficient signal transduction methods. Signal transduction is a process by which a biological signal is transformed and amplified to be recorded. The conventional drugs used for AD can only reduce the progression but cannot be the cure for AD; therefore, currently, the development of a target-based therapy (TBT) is of interest [11,12].

In contrast to conventional treatment of AD, nanocarriers or nano-formulations have numerous advantages, such as bypassing hepatic metabolism, reduced dose, enhanced stability of drugs, and bioavailability and targeted delivery at the site of action. Delivery of drugs to the brain is the most challenging part of treating neurological disorders. The molecular movement in the brain parenchyma is obstructed by a physical interface, the blood–brain barrier (BBB), present between the central nervous system (CNS) and peripheral circulation [13]. It is one of the most challenging physiological barriers. Lipophilic molecules smaller than 400 Da can cross the BBB. A restriction exists for the movement of lipid-insoluble molecules or larger hydrophilic molecules [14,15]. Liposomes, solid lipid nanoparticles (SNPs), gold NPs, and polymeric NPs (PNPs) are some approaches to increase drug movement across the BBB [16,17,18]. These possible substitutes can improve the delivery of the drug across the brain, i.e., drug-loaded PNPs. The high loading capacity of drugs, stability, drug encapsulation, and controlled drug release kinetics are the properties of the PNPs, which can be smoothly enhanced by a range of ligands attached to the NPs surface [19,20]. The specific ligands binding to the surface of NPs and enhancing the passing of restricted therapeutic compounds to the BBB make the drug delivery to the CNS more targeted. For example, polyethene glycol (PEG) can improve the plasma residence time of the NPs by protecting them from mononuclear phagocytes. This keeps NPs in circulation for a longer duration, which increases the possibility of successful organ-targeted delivery across the BBB [21,22,23]. The development of nanotechnology has facilitated the advanced treatment of AD. It has excellent potential for improving the brain-targeting of drugs [23]. The nanoscale drug delivery system consists of PNPs, SLNs, nano-emulsions (NEs), nanostructured lipid carriers (NLCs), dendrimers, and nano-capsules. This nanoscale system can be administered intranasally, transdermal, or intravenously, so that the maximum amount of the drug can reach the brain.

## 2. Conventional Therapeutic Approaches and Their Limitations

The conventional therapeutic approaches consist of non-pharmacological treatment and pharmacological treatment (also referred to as symptomatic or neuroprotective). Symptomatic treatment consists of the N-methyl-D-aspartate (NMDA) receptor antagonist and cholinesterase inhibitors (CHEIs), based on their ability to reduce symptoms across functional areas and control behavioral and psychological symptoms [20]. The goal of pharmacological treatment for AD is to enhance cholinergic transmission inside the brain. Acetylcholinesterase (AchE) inhibition prevents the breakdown of ACh, increasing Ach levels in synapses, but it is associated with nausea and vomiting as side effects. Therapeutics based on Aβ and τ can also be chosen as a target. Non-pharmacological approaches such as sleep, physical activity, and music therapy can also improve the quality of life of a person suffering from AD [21]. Aduhelm (aducanumab-avwa), Aricept (donepezil hydrochloride), Excelon patch (rivastigmine transdermal system), Namenda (memantine HCl), Namzaric (memantine hydrochloride extended-release + donepezil hydrochloride), and Razadyne (galantamine hydrobromide) are Food and Drug Administration (FDA)-approved drugs indicated for use in AD. Sometimes, the combination of donepezil and memantine is also used as an AD therapeutic [22]. Conventional dosage forms, such as solid and liquid dosage forms, have several limitations, such as first-pass metabolism and unfavorable pharmacokinetics, leading to lower bioavailability and a requirement of high doses of therapy [23]. The oral dosage form must pass through the gastrointestinal tract, through effective absorption, sustained systemic circulation, and cross the BBB to reach the target site [24]. The physicochemical properties, such as solubility, molecular weight, polarity, and the partition coefficient, are essential for a drug to show its therapeutic effect, or its suboptimal responses can lead to drug failure [25]. Furthermore, bioactives such as proteins and peptides have higher metabolism via the oral route, which may lead to their suboptimal therapeutic effects [26]. The above discussions are summarized in Table 1.

## 3. Application of Nanomedicines in the Management of Alzheimer’s

Over the past few years, researchers have also explored how nanomaterials might be used in precision medicine [27]. As the only available treatment for AD cannot cross the BBB, it is limited to relieving symptoms. The numerous advantages of nanotechnology-based therapy suggest that this limitation may eventually be overcome [27]. The FDA has approved a wide range of nanocarriers for drug delivery to treat various disorders. Nanocarriers are used to treat various neurological disorders, including Alzheimer’s disease and brain cancer [28]. Nanomaterials are currently under intense study for their potential to manage AD pathologies. Current methods for treating AD involve using delivery systems based on nanostructures, and these are discussed in detail below. Most of these nanocarriers fall into one of four categories: metallic/non-metallic NPs, organic nanoparticles, lipid vesicular, and emulsion-based. Furthermore, various factors cause AD, and a nanotechnological approach is illustrated in Figure 1.

Metallic nanoparticles have been demonstrated as a useful therapeutic approach in managing AD via targeted drug delivery. This includes gold, silver, selenium, iron, and cerium, and they are known to have good anti-AD properties [29]. Gold nanoparticles are assigned for AD because of their transcytosis movement via endothelial cells (without surface modification) of the brain. These positively charged nanoparticles can carry bioactive agents and help in the direction of targeted brain tissues [30]. Gold nanoparticles (Au-NPs) have optimal permeation properties across the BBB as well as neuroprotective properties [30]. Au-NPs in conjugation with glutathione have been discovered for an anti-Alzheimer’s effect as it inhibits Aβ aggregation [31]. Au associated with anthocyanin shows anti-Aβ aggregatory and anti-inflammatory properties. The consumption of Au-NPs decreases AchE levels and offers anti-Alzheimer’s and anti-inflammatory effects [28,30]. An oligomer-specific scFv-AbW20 coupled to superparamagnetic iron oxide NPs (SPIONs) and the class A receptor activator XD4 (W20/XD4-SPIONs) have shown promise in the treatment of AD [32]. SPIONs combined with an A-oligomer-specific antibody and a category A scavenger receptor activator showed exceptional early diagnostic potential for AD [32].

Silica nanoparticles (SiNPs) are used for BBB targeting because of their cellular uptake efficiency and localization in the cytoplasm. These NPs accumulated in the intracellular amyloid cells (Aβ1-42), thus helping to treat AD. The developed SiNPs can decrease cellular apoptosis and reactive oxygen species (ROS) in the intracellular region in a dose-dependent manner. Increased τ phosphorylation and amyloid peptide accumulation are two distinct characteristics of AD. Both were studied for the efficiency of SiNPs in decreasing Aβ1-42 plaque formation and hyperphosphorylation [33]. Intranasal silicon-coated NPs are used for effective brain targeting and are reported in the literature. Reduced levels of ROS in the brain have been identified as a critical component of treating AD. The antioxidant trace elements selenium (II), sodium selenite (VI), and sodium selenite (IV) all play important roles in preventing ROS from damaging cells. Selenium (Se) and selenite are essential micronutrients with antioxidant properties. Se-NPs and selenite NPs have shown increased antioxidant effects. Hence, they can be further studied for potential anti-AD effects. Se-NPs, in combination with sialic acid, increase the permeation through the BBB and reduce the aggregation of Aβ. Hence, this is an effective tool for treating AD [34,35]. Coating sialic acid, peptide-B6, and epigallocatechin-3-gallate (EGCG) with modified selenium NPs possesses high BBB permeability, which may also prevent A aggregation [36]. Newly modified selenium NPs entrapped in poly lactic-co-glycolic acid (PLGA) nanospheres were used for the loading of curcumin and showed potent inhibitory impacts against A aggregation in a transgenic AD mouse model, making it an attractive delivery system for the treatment of AD [37].

Cerium nanoparticles (Ce-NPs) also possess neuroprotective potential and antioxidant effects. They quickly permeate through the BBB and possess no neurotoxic effect. Ce-NPs coupled with triphenylphosphonium (TPP) were studied for AD and were shown to prevent neuronal death and have an anti-Alzheimer’s effect [38].

Solid lipid nanoparticles (SLNs) are spherical nanocarriers that contain a solid lipid core matrix and are the most preferred system for drug delivery to penetrate the BBB, which can solubilize lipophilic molecules [39]. The lipid core comprises monoglycerides, diglycerides, or triglycerides, such as tristearin, glyceryl behenate, and glycerol monostearate [40]. It also consists of fatty acids, steroids, or waxes, surfactants which can stabilize with emulsifiers, preventing particle agglomeration [40]. Different methods, such as solvent evaporation, the solvent emulsification–diffusion method, the ultrasonication/high-shear technique, the precipitation technique, high-pressure homogenization, and the spray-drying method, can develop SLNs [41]. Donepezil (DPL) is available for oral delivery in tablets or capsules (5 or 10 mg/day). These formulations, however, provide non-targeted delivery, which can lead to unwanted effects in the digestive tract (such as diarrhea, nausea, anorexia, and gastric bleeding) and muscle convulsions [15]. DPL is hydrophilic (freely soluble in water), which limits its ability to enter the brain and necessitates frequent dosing, which in turn causes severe cholinergic side effects [42]. According to the prior studies, DPL exhibits hepatotoxicity (though to a lesser extent than its processor tacrine) [42] and undergoes first-pass metabolism, again a limitation for oral delivery [41]. Therefore, it will be helpful to develop a non-oral delivery system of DPL to reduce the risks associated with oral delivery and avoid systemic drug exposure and off-target drug distribution. Since the brain is where DPL is most effective, any newly developed system must ensure that the therapeutic concentration of the drug is directly delivered to the brain. Due to this, SLNs (which contain DPL) will be administered via the intranasal route [43].

Furthermore, the optimized SLN was found at 121 nm with a zeta potential of −24.1 mV and an entrapment efficiency of 67.95 percent [43]. Albino Wistar rats were used for the pharmacokinetic and biodistribution studies of intranasal DPL-SLNs. The authors found an AUC 2.61 times higher than the intravenously (i.v.) administered DPL-Sol and 2.26 times higher than the intranasally administered DPL-Sol. The scintigraphy study of the rabbit’s brain showed the drug’s location [43]. Therefore, it was concluded that the optimal DPL-SLN had a shelf-life of 2.29 years. As a result, the study proved that SLNs are helpful in the delivery of DPL to the brain. Similarly, curcumin-loaded SLNs were prepared and studied. The study shows that they reduced the behavioral dysfunction and reversed several neurotransmitters in the brain [44]. Pomegranate extract-loaded lipidic nanoparticles (LNPs) were also studied in an aluminum chloride-induced rat model of AD. Pomegranate extract has tannins and alkaloids, which show high antioxidant effects. It decreased neurofibrillary tangles (NFTs) and beta amyloid (Aβ) deposition [45]. α-Bisabolol is a sesquiterpene alcohol found in *Matricaria chamomilla* and it shows beneficial effects, such as anti-cholinesterase, anti-plasmodial, anti-inflammatory, anti-cancer, and antimicrobial, but it has low solubility and bioavailability [46]. α-Bisabolol-loaded cholesterol LNPs prevent neuro-2α cells from Aβ-induced neurotoxicity and inhibit Aβ aggregation [46]. Erythropoietin (EPO) helps in neuronal survival and regulates AD. However, it has decreased penetration through the BBB due to its hydrophilicity, high molecular weight, and rapid clearance from the bloodstream. EPO-encapsulated SLNs decrease oxidative stress and Aβ deposition and show increased spatial memory [47].

Nanostructured lipid carriers (NLCs) are another promising strategy for targeted drug delivery, consisting of a disorganized inner lipid matrix of solid and liquid lipids. NLCs can imitate the natural lipid environment of bio-membranes, including the BBB. NLCs exhibit high affinity to Aβ and promote its degradation [48]. NLCs are more stable and have a good loading capacity [49]. NLCs are used for the delivery of drugs from the nose to the brain and in AD treatment associated with microglial activation [50]. The NLCs have merits over conventional dosages because of their lipophilic nature and smaller size compared to the particle size of conventional dosage forms, which helps the drug molecules easily cross the BBB [51]. The lipid matrix protects the NLCs from being degraded by enzymes and allows the active drug to reach the target site [52,53]. The main limitation of NLCs is their physical instability and safety, which can be avoided by modifying the temperature, storage, and pH [54,55]. Curcumin-loaded NLCs treat oxidative stress conditions in AD. They increase the curcumin bioavailability in the brain and reduce the hallmark of Aβ in AD [56,57]. Lipid nano-capsules (LNCs) with a lipid core and loaded with indomethacin (Ind) have been investigated for their potential to prevent Aβ1-42-induced cell damage and neuroinflammation. This demonstrated that Ind-LNCs inhibit neuroinflammation induced by Aβ1-42 in organotypic hippocampal cultures and decrease A-induced cell death [57].

Polymeric nanoparticles (PNPs)/polymeric nanomaterials have a wide variety of structures and morphologies and range in size from just a few nanometers to over 1000 nm. This novel material is derived from nanotechnology, with the potential to revolutionize drug delivery and other biomedical fields. Both natural and synthetic polymers can be used in the production of PNPs, with each method yielding its own unique set of characteristics. PLGA is a polymer that provides targeted brain delivery and increased NP uptake. Surface functionalization of PLGA-NPs increases their transport through the BBB, aiding in AD treatment. Besides loading the drug in the polymer matrix core of nanoparticles, PLGA improves drug safety against degradation. It propounds the probability of altering the in vivo and in vitro release profiles [58]. A study has been conducted on hydrophilic PLGA-coated curcumin NPs in conjugation with Tet-1 peptide. Curcumin is a phytoactive molecule. Researchers developed PLGA-coated curcumin NPs, showing an antioxidative effect and demolishing Aβ aggregates in the AD animal model [59]. A study on vitamin D for its neuroprotective effect has been carried out, but it has low solubility and poor bioavailability. The murine AD model was used to study vitamin D-loaded PLGA-NPs. Decreased neuronal apoptosis with neuroinflammation and increased cognitive function were observed [60]. Huperzine A was loaded into PLGA conjugated with lactoferrin NPs, showing enhanced release kinetics and significantly decreased AD symptoms [61]. Selenium NPs encapsulated into PLGA nanospheres with curcumin show strong inhibition against Aβ aggregation and can be used as a targeted drug delivery method in treating AD [37]. Memantine is an FDA-approved drug used in the treatment of AD. Polymer-based NPs (PBNPs) associated with memantine show anti-inflammatory and anti-Alzheimer’s effects [62]. Zinc and sitagliptin-loaded NPs show improved cognitive dysfunction and reduced neuroinflammation as anti-Alzheimer’s effects [63]. Chitosan NPs, in combination with rivastigmine by emulsification, have been studied to treat AD. Through the zeta potential of NPs by coating with polysorbate 80, it was observed that the positive charge of chitosan NPs was reduced. Immuno-vehicles were designed for chitosan-coated PLGA-NPs conjugated with a novel anti-Aβ antibody. They showed increased BBB permeation and better drug targeting. In addition, chitosan increases the aqueous dispersibility and stabilizes immune nano-vehicles during lyophilization [64]. PEGylated polylactic acid was conjugated with TGN and QSH (two targeting peptides) to develop a NP drug delivery system that selectively targets BBB ligands and Aβ1-42. These NPs precisely target AD lesions without having observable cytotoxicity [34].

Thymoquinone (TQ) is a bioactive component found in the essential oil of *Nigella sativa* seeds and has a wide range of therapeutic uses [35]. Many preliminary pharmacological studies have been performed to examine TQ’s therapeutic use. TQ has been shown in recent studies to be an effective treatment for AD [65]. TQ-containing PLGA-NPs with polysorbate-80 (P-80) could safely and effectively transport NPs across the BBB and into the brain [66]. The PLGA-NPs are shielded from being opsonized and cleared by the body because of the P-80 surfactant coating. P-80 is a non-toxic, non-ionic, biodegradable, and hydrophilic surfactant [67]. TQ from P-80-TQNPs was easily derived due to the hydrophilicity of the P-80 coating [67]. The amount of TQ delivered may have been limited due to the interaction between TQ and PLGA, which caused a further delayed discharge. TQ reduces the production of superoxide radicals primarily by blocking the enzyme xanthine-oxidase. A summary of the above discussion is presented in Table 2.

Dendrimers are a class of compounds that have shown promise for use in treating AD [68]. Dendrimers are highly branched polymeric materials with a core, branches, and terminal functional groups with a unique architecture. Dendrimer scaffolds have a central core comprising atoms or molecules with at least two identical chemical functional groups. Dendrimers with an ethylenediamine core, particularly those of the 4th and 5th generations, are commonly used for CNS disorders such as AD. The discovery was made by targeting memantine delivery to specific brain regions in AD-induced mice using a combination of low-generation dendrimers and lactoferrin. In recent studies, an important impact on the memory aspects of the target mice was observed [80].

Dendrimers with a poly (propylene imine) core and a maltose histidine shell (G4HisMal) have been successfully developed, and they may show significant alleviation of AD symptoms, such as memory dysfunction. Furthermore, due to their improved solubility, bioavailability, and ability to permeate the BBB, they can more precisely target the damaged parts of the brain. Together with generation 4.0 and PAMAM dendrimers as nanocomposites, tacrine has been employed [68]. This has improved the biocompatibility and reduced the toxicity of the drugs used to treat AD.

Liposomes are lipid vesicular nanocarriers, which are composed of cholesterol and a wide variety of phospholipids (such as phosphatidylcholine, phosphatidylethanolamine, phosphatidylserine, and phosphatidylglycerol). They have one or more lipid bilayers, ranging in size from about 20 nm to about 1000 nm [80]. Self-mobilizing lipid nanocarriers integrate and supply the drug into the CNS with amphiphilic properties [81]. With peptides that penetrate cells and proteins, liposomes offer a surface that may be successfully modulated [82]. Thus, this has provided methods for targeted and site-specific drug delivery and the passage of drug molecules through the BBB [83]. Curcumin-loaded liposomes can deliver the drug to the CNS, permeate the BBB, and show an anti-Alzheimer’s effect [84]. Osthole (a derivative of coumarin) is known for its preventive effect against hippocampal neuron [69] compliance as it reduces the frequency of administration, and it can be used as a treatment for AD [70]. Donepezil via oral drug delivery cannot cross the BBB. Donepezil-loaded liposomes are made and intranasally administered to rapidly cross the BBB, showing improved bioavailability and reducing systemic toxicity [85].

Ethosome is a novel liposomal delivery system containing small vesicles with a high ethanol concentration (45 percent *w*/*w*), water, and phospholipids [86]. Its size can vary from 10 nm to 1000 nm [87]. It can deliver to and permeate across the skin’s deep layers, and hence is widely used in transdermal drug delivery. In contrast with liposomes, ethosome has a negative superficial charge because of ethanol inclusion [86]. Ligustrazine phosphate (LP) has been studied to treat AD via antioxidant action [71]. LP is a hydrophilic molecule with less skin permeability to achieve therapeutic blood concentrations. When comparing the LP ethosomal system and the aqueous solution, the ethosomal system proved better [71]. Ethosome can overcome this limitation as it has an enhanced permeation carrier, which helps in skin permeability [86]. These characteristics of ethosome can be helpful in BBB penetration.

Regarding micro/nano-emulsion-based nanocarriers, microemulsion (ME) is a method by which the BBB can be bypassed, and the drug can be delivered directly to the brain using the nasal olfactory route [88]. ME helps in quick drug absorption via the highly vascularized mucosa, which improves the brain’s drug concentration and helps deliver drugs to a specific brain site [72]. Morin hydrate is an antioxidant drug that helps treat AD by inhibiting extracellular accumulation of Aβ peptide and intracellular neurofibrillary tangles, the two major neuropathological features in AD [89]. The simple morin aqueous solution administered via the parenteral route has several drawbacks, such as safety issues, low patient compliance, and expensive medication. Avoiding the BBB, intranasal delivery of morin hydrate-loaded ME is a potential strategy, and it offers an advantage as it is non-invasive. The ME is an effective option to overcome low aqueous solubility. ME has several advantages, such as it increases drug solubilization, offering absorption across mucosal membranes, and has a high drug loading capacity [90]. ME increases the drug bioavailability by increasing the plasma half-life, thus increasing the circulation time, which increases the contact time of the drug with the BBB [91].

Researchers found that people who took NSAIDs such as ibuprofen had a lower risk of developing AD [92]. A novel repurposing strategy and route of administration are presented in this study for the treatment of AD. The in vivo result in rats found uptake of a novel ibuprofen-loaded ME nearly four times higher than that of the intravenous and 10 times that of the oral administrations [73]. The ME loaded with tacrine was more effective than intranasal administration of the drug solution regarding absorption. The intranasal administration of tacrine-loaded ME resulted in the quickest memory recovery in scopolamine-induced amnesic mice [74]. Huperzine A-loaded ME improves cognitive function in mice compared to oral suspension. A ME-based patch was developed to deliver huperzine A and ligustrazine phosphate through the transdermal route. This combination therapy has a synergistic effect against amnesia [93].

Nano-emulsion is another emulsion-based nanocarrier system that is used to deliver the drug in AD. Nano-emulsion increases the efficacy of the anti-AD drug and targeted drug delivery in AD [94]. Naringenin is a natural bioactive molecule known for its anti-apoptotic, anti-inflammatory, antioxidant, and anti-AD effects. However, it has limitations of lower solubility and poor bioavailability. A nano-emulsion of naringenin was evaluated for its anti-AD effect. The study outcome shows that the nano-emulsion of naringenin could be used to overcome Aβ neurotoxicity and amyloid genesis [76]. Memantine is an FDA-approved therapy, and it reduces the progress of AD. A nano-emulsion of memantine using homogenization and ultrasonication was studied for its anti-Alzheimer’s effect. It was administered via the intranasal route. This nano-emulsion crosses the BBB and increases the anti-AD effect compared with the conventional dosage form [77]. Similar to memantine, a cubosomal-loaded donepezil mucoadhesive was formed and tested for targeted delivery in treating AD [95]. Both in vitro and in vivo experiments showed promising results of emulsions in AD pathologies [77]. A nano-emulsion containing DPL hydrochloride was developed using labrasol and glycerol at 10 percent w/w. The DPL hydrochloride nano-emulsion can potentially treat AD, due to its antioxidant and radical scavenging effects [78]. The effects of a lactoferrin-modified nano-emulsion on an in vitro brain model were examined. Lactoferrin acts as an iron-binding lactoferrin, which is a glycoprotein [96]. It demonstrates brain-targeted delivery systems and improved drug release in the brain parenchyma [78]. Thus, intranasal administration of the developed nano-emulsion was targeted to brain cells and tissues. In a recent study, deferoxamine delivered via nanogels made of chitosan and tripolyphosphate was shown to be an effective treatment for AD [79]. The above-discussed nanocarriers are summarized in Table 2. 

## 4. Conclusions

In recent decades, numerous efforts have been made to treat neurodegenerative disorders such as AD. Current treatment of Alzheimer’s disease includes FDA-approved drugs such as galantamine, donepezil, rivastigmine, and memantine, administered orally. Due to physiological factors and disease complexity, the conventional therapy approach has several drawbacks. Despite several research investigations, the current AD treatment is ineffective due to the gaps between promising research findings and appropriate clinical trial implementation. Recent advancements in nanotechnology appear to provide innovative ways to boost research via targeted drug delivery at a brain-targeted specific site. Mitochondrial dysfunction may be a key focus for treating AD. The clinical challenge of AD treatment is also linked with various side effects of drug therapy. As an alternative, nanomedicines are known to be the best option for the treatment of AD as compared to conventional therapeutics. The immunogenicity, biocompatibility, high biological affinity, particle size, biodegradability, and low toxicity are responsible for the high therapeutic potential of nanocarriers. Nanocarriers effectively deliver hydrophobic as well as hydrophilic drug molecules through the BBB. The recent developments in nanomedicine and nanotechnology for treating AD have extensively assigned the current state of nanocarriers for anti-Alzheimer’s drug delivery. However, more potent, advanced, and non-toxic nanomedicine formulations are required to overcome the challenges presented by AD. Advancing these nanocarriers from research to the clinic is the real challenge. The clinical failures may be due to the possible toxicity and slow degradability with polymeric nanoparticles. Without surface modifications, the ability of the polymeric NPs to cross the BBB is restricted. Lipid nanoparticles have poor loading of hydrophilic drugs, and liposomes have poor in vivo stability. Upon storage, nano-emulsions show poor stability, resulting in phase separation and an immediate release effect. Dendrimers also have toxicity issues. Therefore, further extensive research studies are required to optimize all such issues and address them in the near future.

## 5. Future Prospective

Although the mechanism responsible for the pathophysiology of AD is unknown, nanotechnology offers a better alternative treatment option for managing AD. Therefore, there is an urgent need to investigate a nanotechnology-based approach for treating AD in preclinical and clinical trials. The toxicity of nanomedicines is another major challenge, which can be conquered by developing more biocompatible nanocarriers. Until now, no nanocarriers has been available in the market, despite much research to bring AD therapy from the research stage to the clinic. However, as nanotechnology has only been investigated in preclinical trials, further research is still needed to enhance it from pharmacokinetic and toxicological standpoints to pass clinical trials. Magnetic nanoparticles are currently unexplored, and dendrimers and nano-emulsions also need to be further explored.

## Figures and Tables

**Figure 1 biomedicines-11-01752-f001:**
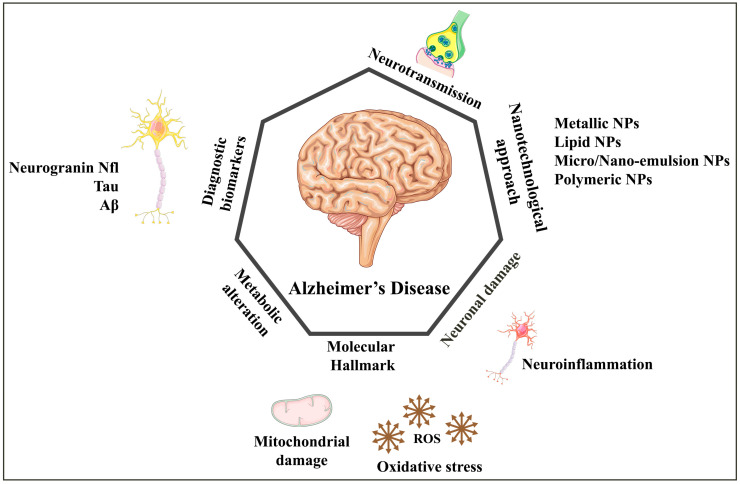
Causative factors of Alzheimer’s disease, targeting agents, and use of various nanocarriers in the managements of Alzheimer’s disease.

**Table 1 biomedicines-11-01752-t001:** Alzheimer’s disease: pathology, risk factors, blood–brain barrier, and therapeutic approaches.

Topic	Information	References
Alzheimer’s disease (AD)	Progressive neurological condition that causes irreversible dementia syndrome. Incidence increases with age: currently affects 32.6 million people worldwide. Expected to reach 78 million by 2030 and 139 million by 2050 without new treatments.	[1,4]
Pathology of AD	Caused by extracellular aggregation of amyloid beta (Aβ) plaques and accumulation of intraneuronal neurofibrillary tangles of tau (τ) protein	[7]
Risk factors for AD	Associated with cardiovascular risk factors such as hypertension, diabetes, atherogenic dyslipidaemia, and obesity—environmental exposure to toxicants, genetic factors, mutation, trauma, and metabolic diseases can also cause AD.	[8,9]
Blood–brain barrier (BBB)	It is one of the most challenging physiological barriers. The drug movement in the brain parenchyma is obstructed by a physical interface, the BBB, present between the central nervous system (CNS) and peripheral circulation. Lipophilic molecules smaller than 400 Da can cross the BBB.	[12]
Conventional treatment of AD	Non-pharmacological and pharmacological approaches: Non-pharmacological approaches such as sleep, physical activity, and music therapy can also help. Symptomatic treatment with NMDA receptor antagonists and cholinesterase inhibitors (CHEIs) are common pharmacological approaches.	[20]
Food and Drug Administration (FDA)-approved drugs for AD	Aduhelm (aducanumab-avwa), Aricept (donepezil hydrochloride), Excelon patch (rivastigmine transdermal system), Namenda (memantine HCl), Namzaric (memantine hydrochloride extended-release + donepezil hydrochloride), and Razadyne (galantamine hydrobromide).	[21,22]
Current treatments for AD	Only reduce symptoms. Temporarily enhance cognitive abilities. Lack brain specialization and cause adverse effects.	[22]
Limitations of conventional treatment	First-pass metabolism and unfavorable pharmacokinetics—lower bioavailability and high dosage requirements. Physicochemical properties can affect drug effectiveness: bioactives may have suboptimal therapeutic effects via the oral route.	[23,24]
Nanotechnology and AD	Nanotechnology can aid in early detection and drug development for AD. Nanocarriers have advantages over conventional treatment. Drug-loaded nanocarriers can improve drug delivery to the brain across the BBB. Further details are described in Table 2.	[25]

**Table 2 biomedicines-11-01752-t002:** Various drug-loaded nanocarriers in the management of Alzheimer’s disease.

Drugs	Nanocarriers	Outcomes/Benefits	References
DPL	Liposome	Donepezil via oral drug delivery cannot cross BBB. Donepezil-loaded liposome are made and intranasally administered to rapidly cross the BBB and shows improved bioavailability and reduces the systemic toxicity of it against AD.	[15]
A oligomer-specific scFv-AbW20	superparamagnetic iron oxide NPs (SPIONs)	This study found have promise results against AD and exceptional early diagnostic potential.	[32]
Silicon dioxide	Silicon nanoparticle	Significantly induced cellular apoptosis, elevate the level of intracellular ROS in dose-dependent manner, decrease the cell viability, enhanced phosphorylation of tau at Ser262 and Ser396.	[33]
Sialic acid	Selenium (Se)-NPs	The loading of Sialic acid into Se-NPs increase the permeation through BBB and reduces the aggregation of Aβ in the animal model of AD.	[36]
Curcumin	Selenium NPs encapsulated into PLGA nanospheres	This study found strong inhibition against Aβ aggregation and can be used as targeted drug delivery in treating AD.	[37]
Triphenyl phosphonium	Cerium nanoparticles	TPP-ceria NPs effectively penetrate mitochondria to scavenge ROS to reduce oxidative stress.	[38]
Donepezil (DPL)	SLNs	Intranasal administration of DP-SLNs significantly increase the concentration of drug in brain over the i.v. administration of DPL solution. Further, the scintigraphy study observed localization of DPL-SLNs into the rabbit’s brain.	[43]
Pomegranate extract	LNPs	This study found higher antioxidant effects and decreased NFTs and Aβ deposition in the aluminium chloride-induced rat model of AD.	[45]
α-Bisabolol	LNPs	Protect the neuro-2a cells from inhibited Aβ aggregation and Aβ induced neurotoxicity	[46]
Erythropoietin (EPO)	Solid lipid nanoparticles (SLNs)	Erythropoietin (EPO) helps neuronal survival and regulates AD, but very limited BBB permeation, due to hydrophilicity and high molecular weight and rapid clearance from the blood stream. EPO-encapsulated SLNs overcomes abovementioned issues and decrease oxidative stress and Aβ deposition and show increased spatial memory.	[47]
Curcumin	NLCs	This study found enhanced curcumin bioavailability in the brain and reduces the hallmark of Aβ in AD	[51]
Curcumin	SLNs	The study found to reduces the behavioural dysfunction and reverses several neurotransmitters into the brain against animal model of AD.	[55]
Curcumin	Liposomes	It can deliver the drug to CNS, permeate the BBB, and show better anti-Alzheimer’s effect in animal model.	[56]
Indomethacin (Ind)	Lipid nano capsules (LNCs)	This study has been investigated that Ind loaded LNCs inhibit neuroinflammation induced by Aβ1-42 in organotypic hippocampal cultures and decrease A-induced cell death in AD animal model	[57]
Vitamin D	PLGA-NPs	Vitamin D observed neuroprotective effect, but poor solubility and bioavailability. The vitamin D loaded PLGA-NPs studied on murine AD model, results to decreased neuronal apoptosis and enhanced cognitive function was observed.	[60]
Huperzine A	PLGA-NPs	Huperzine A was loaded into PLGA-NP conjugated with lactoferrin, showed enhanced release kinetics, and significantly decreased AD symptoms.	[61]
Memantine	Polymer-based NPs (PBNPs)	Memantine loaded PBNPs shows effective an anti-inflammatory and anti-Alzheimer’s effect against AD animal model.	[62]
Zinc and sitagliptin	PBNPs	It shows improved cognitive dysfunction and reduced neuroinflammation when studied for their anti-Alzheimer effect against AD animal model.	[63]
Thymoquinone (TQ)	PLGA-NPs	TQ-containing PLGA NPs with polysorbate-80 (P-80) could be a safe and effective way to transport NPs across the BBB and into the brain. The PLGA-NPs are shielded from being opsonized and cleared by the body because of the P-80 surfactant coating. TQ works by reducing the production of superoxide radicals primarily through blocking the enzyme xanthine-oxidase.	[66]
Tacrine	Dendrimers with a poly (propylene imine) core and a maltose histidine shell (G4HisMal)	Tacrine loaded into generation 4.0 and PAMAM dendrimers has been employed and has improved biocompatibility and reduced the toxicity of the drug used to treat the AD.	[68]
Osthole (Coumarin derivative)	Liposome	This study found increased intracellular uptake by APP-SH-SY5Y cells and exerted a cytoprotective effect. Prolonged the cycle time and elevate the accumulation of Osthole in the brain	[69]
Rivastigmine	Liposome	Increase the concentration and exposure in the brain.	[70]
Ligustrazine phosphate	Ethosome	Drug penetration and deposition significantly higher over the plain drug.	[71]
Morin hydrate	ME	The morin hydrate solution given by a parenteral route has several drawbacks, such as safety issues, low patient compliance, and expensive medication. Avoiding the BBB, intranasal delivery of morin hydrate-loaded ME is a potential strategy, and it offers an advantage as it is non-invasive.	[72]
Ibuprofen	Microemulsion (ME)	A novel repurposing strategy and route of administration are presented in this study for the treatment of AD. The in vivo result in rats found uptake of a novel ibuprofen loaded ME nearly four times higher than that of the intravenous and ten times than that of the oral administrations.	[73]
Tacrine	ME	The intranasal administration of tacrine-loaded ME results in the quickest memory recovery in scopolamine-induced amnesic mice.	[74]
Huperzine A	ME	Huperzine A loaded ME improves cognitive function in mice compared to oral suspension.	[75]
Naringenin	Nanoemulsion	The study outcome shows that nanoemulsion of naringenin could be used to overcome Aβ neurotoxicity and amyloid genesis.	[76]
Memantine	Nanoemulsion	Memantine loaded nanoemulsion using homogenization and ultrasonication was studied for its anti-Alzheimer effect. It was given by intranasal route. This nanoemulsion crosses the BBB and increases the anti-AD effect compared with the conventional dosage form.	[77]
DPL	Nanoemulsion	Using labrasol and glycerol at a concentration of 10% w/w, a nanoemulsion containing donepezil hydrochloride was developed. Donepezil hydrochloride nanoemulsion has the potential to treat AD, due to its antioxidant and radical scavenging effects.	[78]
Deferoxamine	Nanoemulsion gel	Deferoxamine delivered via nanogels made of chitosan and tripolyphosphate shows an effective therapeutic action against AD.	[79]

LNPs: lipid nanoparticles; PBNPs: polymer-based NPs; NLCs: nanostructured lipid carriers; SLNs: solid lipid nanoparticles; BBB: blood–brain barrier; AD: Alzheimer’s disease; PLGA: poly D,L-lactic-co-glycolic acid; DPL: donepezil.

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
