# Peer review of "Nanomedicine in the Management of Alzheimer’s Disease: State-of-the-Art"

_biomedicines, 2023, doi:10.3390/biomedicines11061752_

Round 1

Reviewer 1 Report

This review summarizes the latest achievements regarding the development of drug delivery systems employing nanotechnology against Alzheimer’s Disease. In spite the fact that there many reviews in the literature focusing in that particular thematic area, thus it is not a scientifically sound work, it is a sincere attempt. The main drawback of the manuscript is the lack of tables that summarize all the relevant information concerning each Section. Table 1 is not adequate and do not include all the provided data.

Author Response

Reviwers Comments 1: 

This review summarizes the latest achievements regarding the development of drug delivery systems employing nanotechnology against Alzheimer’s Disease. In spite the fact that there many reviews in the literature focusing in that particular thematic area, thus it is not a scientifically sound work, it is a sincere attempt. The main drawback of the manuscript is the lack of tables that summarize all the relevant information concerning each Section. Table 1 is not adequate and do not include all the provided data.

Author Response: Dear Reviewer, Thank you for your remarks. Now Table 1 has been added to summarized each section apart from the nanotechnology section. Further, the nanotechnology section has been given in Table 2.

Reviewer 2 Report

Theoretically, the manuscript of Ullah et al. could be a valuable minireview on the treatment of Alzheimer's disease, but the current form is very far from publication quality. Maybe an intensive revision can transform it into acceptable, though the authors need to invest efforts to reach this goal.

The referee's principal concerns touch the Abstract, the manuscript organization, and the currently messy Reference sections.

- The sentences in lines 28-31 are better placed in the Introduction because they belong to the rationale rather than the Abstract.

The sentence in lines 35-37 needs reformulation because the current version is messy.

- In the Abstract, the authors wrote about 35 million affected people (line 27), while in the Introduction (line 44), the reader finds 42.3 million. As these values do not match, please decide which value is more or less correct and stick to that one.

- The referee recommends combining 3.1.1 (Liposome) and 3.1.2 (Ethosomes) sections, as the authors wrote about their minimal differences. On the other hand, as the expression 'ethosomes' is in the plural, the term 'liposomes' is also a summary name, and it is singular. In combination, the sub-subchapter numbering is also useless.

Section 3.3. is in the wrong location and has a better place at the end of Chapter 3, as other NPs are organic NPs. Besides this, the single sub-subchapter has the number 3.3.2 since there is no 3.3.1 section, and this chapter contains only one paragraph.

There is a figure and table at the end of the manuscript, but their referencing is missing, and neither is their location seen.

- The reference section contains full and abbreviated journal names. The referee would recommend googling or using the Academic Accelerator for the correct abbreviations of journals.

Some other minor concerns are summarized below.

- The authors defined beta (what is otherwise a valid Greek letter) in line 48 but omitted it in the case of tau. Tau is also a living Greek letter (τ). The referee recommends using the correct character instead of the English transcription (e.g., in line 102).

- In line 114, the referee is unsure that 'barriers' is the correct word and perhaps 'tract' is better.

- Please combine the two sentences in lines 117-121 because their meaning is practically identical.

- Lines 134-139 contain two sentences of minimal differences. Please rephrase them into one sentence.

- Line 144 informs the reader that ethosomes have small dimensions. As 'small' is relative and the liposomes dimensions are also missing, please provide quantitative information for the lipo- and ethosomes dimensions because the authors' message is pending.

- In line 169, the authors wrote something about the systemic toxicity of donepezil. First, what FDA has approved is unclear: the donepezil or the donepezil-SLN formulation. The second concern is that in the previous section, they wrote that liposomes significantly reduce the cellular toxicity of donepezil. Does the donepezil systemic toxicity correlate with cellular toxicity? Please provide some words to explain the potential (maybe virtual) contradiction.

- Sentence in lines 171-172 (with ref. 44) is a repetition. The authors mentioned donepezil several times, and the manuscript is about the treatment of Alzheimer's disease, so this sentence (not the quote!) is unintelligible because of its obvious content.

- Lines 177-178 contain a repetition in the essential oil - assumed - beneficial effects.

- Lines 185 and 186 contain two kinds of spelling. Please decide which one is correct and use that form in the text.

- In line 193, the authors again compare the sizes of something (smaller size) but have forgotten to inform the reader: comparing to what.

- Lines 206-209 contain a repetition. The reader will understand at first reading what is meant by the limited applicability of metal nanoparticles (="every metallic nanoparticle is not suitable for targeted drug delivery"), despite the grammatical challenges of the message in parentheses.

- As mentioned before, Section 3.4 has a better place after Section 3.2, as the polymeric nanoparticles are organic materials.

- Line 239 contains redundant information, as the whole manuscript is on a neurodegenerative disease.

- Message about the curcumin (lines 243-244) is also repeated there.

- The authors confused the terms in the sentence about PLGA nanoparticles (lines 244-245). Not curcumin encapsulates PLGA, but vice versa. It is clear from the title of the cited article. Have the authors read that publication?

- In line 257, it is again unclear whether the FDA has approved the memantine API or the nanoformulation.

- In line 261, the authors discuss the emulsified chitosan/rivastigmine formulation. Why is it here and not in the next chapter, which discusses the emulsion-based nanocarriers?

- In lines 287-288, it is unclear whether the AD-reducing property is a general character of the NSAIDs or the NSAIDs nanoformulations have this property only.

- The name of compounds, except brand or specific cases, should not start with a capital letter inside a sentence (e.g., naringenin, NOT Naringenin).

- The existence of Chapter 4 is questioned. The content of this section is effectively one of the concluding remarks and therefore fits better into the Conclusion. Generally, the title of the chapters in a manuscript is a recommendation and not a strict rule - except, of course, for a few necessary chapters, like the Introduction, Discussion, or Conclusion.

- In line 321, the administered sounds better than the given.

- Lines 346-351 are in the wrong location. These sentences are more appropriate in the Conclusions section, as they are not consistent with the title of the chapter.

- As mentioned before, some journal abbreviations are missing (refs. 3, 7, 8, 9, 10, 12, 18, 19, 25, 26, 27, 30, 32, 33, 42, 43 - wrong punctuation mark -, 45, 51, 52, 55, 60, 62, 68, 70, 77, 81, 86). In case of doubts, please use Google or any other search engine. Please also note that if the journal name end is not an abbreviation, a dot is unnecessary between the name and the year. For example, refs 1 and 13 are correct, while ref 2 is not.

Reformulation of some sentences is necessary, principally due to the redundant or repeated content. The manuscript contains many very long sentences which require modifications, too.

Author Response

Reviwers Comments 2:

Theoretically, the manuscript of Ullah et al. could be a valuable minireview on the treatment of Alzheimer's disease, but the current form is very far from publication quality. Maybe an intensive revision can transform it into acceptable, though the authors need to invest efforts to reach this goal.

Author Response: Dear Reviewer, thank you for your remarks. We tried are best to revise extensively and hope it can be transformed into acceptance now.

The referee's principal concerns touch the Abstract, the manuscript organization, and the currently messy Reference sections.

Author Response: Now, revised these section as per the suggestions.

- The sentences in lines 28-31 are better placed in the Introduction because they belong to the rationale rather than the Abstract.

Author Response: Dear Reviewer, Thank you for your worthy suggestions. Accordingly placed in the introduction section as shown in red colour text.

The sentence in lines 35-37 needs reformulation because the current version is messy.

Author Response: Dear Reviewer, thank you for your remarks. Accordingly, it has been revised in the abstract section (shown in red colour text)

In the Abstract, the authors wrote about 35 million affected people (line 27), while in the Introduction (line 44), the reader finds 42.3 million. As these values do not match, please decide which value is more or less correct and stick to that one.

 Author Response: Dear Reviewer, thank you for remark. It has been corrected now and placed with the reference. At the both places, the same content as given below.

It currently affects 32.6 million people worldwide, but without new treatments, it is expected to nearly double every 20 years, reaching 78 million by 2030 and 139 million by 2050 [4].

The referee recommends combining 3.1.1 (Liposome) and 3.1.2 (Ethosomes) sections, as the authors wrote about their minimal differences. On the other hand, as the expression 'ethosomes' is in the plural, the term 'liposomes' is also a summary name, and it is singular. In combination, the sub-subchapter numbering is also useless.

Author Response: Dear Reviewer, Thank you for worthy suggestions. Now the different section of nanocarriers is merged and now it come under “Application of nanocarriers in the management of Alzheimer's.” On the other hand, expression of ethosomes and liposomes has been corrected.

Section 3.3. is in the wrong location and has a better place at the end of Chapter 3, as other NPs are organic NPs. Besides this, the single sub-subchapter has the number 3.3.2 since there is no 3.3.1 section, and this chapter contains only one paragraph.

Author Response: Dear Reviewer, thank you for worthy suggestions. Now, all the subsection of chapter 3 has been merged now. As per the given suggestions, nanoparticles are placed.

There is a figure and table at the end of the manuscript, but their referencing is missing, and neither is their location seen.

Author Response: Figure is original. In the table, now proper references have been cited. Now, figure and table has been placed at the right location in the text only.

The reference section contains full and abbreviated journal names. The referee would recommend googling or using the Academic Accelerator for the correct abbreviations of journals.

 Author Response: Dear reviewer thank you for remarks. Now, all the reference has been checked carefully and corrected now for the above-mentioned points.

Some other minor concerns are summarized below.

- The authors defined beta (what is otherwise a valid Greek letter) in line 48 but omitted it in the case of tau. Tau is also a living Greek letter (τ). The referee recommends using the correct character instead of the English transcription (e.g., in line 102).

 Author Response: Dear Reviewer, thank you for your remarks. Now, it has been corrected throughout in the text.

- In line 114, the referee is unsure that 'barriers' is the correct word and perhaps 'tract' is better.

 Author Response:  Correction have been made.

- Please combine the two sentences in lines 117-121 because their meaning is practically identical.

 Author Response: Corrections have been made

- Lines 134-139 contain two sentences of minimal differences. Please rephrase them into one sentence.

Author Response: Dear Reviewer, Thank you for suggestions. The sentences have been revised now.

Donepezil via oral drug delivery cannot cross BBB. Donepezil-loaded liposomes are made and intranasally administered to rapidly cross the BBB, showing improved bioavailability and reducing systemic toxicity [76].

- Line 144 informs the reader that ethosomes have small dimensions. As 'small' is relative and the liposomes dimensions are also missing, please provide quantitative information for the lipo- and ethosomes dimensions because the authors' message is pending.

 Author Response: Dear Reviewer, Thank you for worthy points. Now, quantitative and their size range has been provided for liposome and ethosome.

Liposome is a lipid vesicular nanocarrier, which is composed of cholesterol and a wide variety of phospholipids (such as phosphatidylcholine, phosphatidylethanolamine, phosphatidylserine, and phosphatidylglycerol). It has one or more lipid bilayers, ranging in size from about 20 nm to about 1000 nm [69].  

Ethosome is a novel liposomal delivery system containing small vesicles with high ethanol concentrations (45 percent w/w), water, and phospholipids [77]. Its size can vary from 10 nm to 1000 nm [78].

- In line 169, the authors wrote something about the systemic toxicity of donepezil. First, what FDA has approved is unclear: the donepezil or the donepezil-SLN formulation. The second concern is that in the previous section, they wrote that liposomes significantly reduce the cellular toxicity of donepezil. Does the donepezil systemic toxicity correlate with cellular toxicity? Please provide some words to explain the potential (maybe virtual) contradiction.

 Author Response: Dear Reviewer, thank you for remarks.  Donepezil (DPL) is a FDA approved drug used in the management of AD. DPL-SLN formulation is not FDA approved. Due to systemic drug exposure and off-target drug distribution of DPL by oral route, lead to systemic side effects and it is not cellular toxicity. Further it is explained below.

Donepezil (DPL) is available for oral delivery in tablets or capsules (5 or 10 mg/day). These formulations, however, provide non-targeted delivery, which can lead to unwanted effects in the digestive tract (such as diarrhea, nausea, anorexia, gastric bleeding, and muscle convulsions.) [15]. DPL is hydrophilic (freely soluble in water), which limits its ability to enter the brain and necessitates frequent dosing, which in turn causes severe cholinergic side effects [42]. According to the prior studies, DPL exhibited hepatotoxicity (though to a lesser extent than its processor tacrine) [42] and undergoes first-pass metabolism, again a limitation for oral delivery [41]. Therefore, it will be helpful to develop a non-oral delivery system of DPL to reduce the risks associated with oral delivery and avoid systemic drug exposure and off-target drug distribution. Since the brain is where DPL is most effective, any newly developed system must ensure that the therapeutic concentration of the drug is delivered directly to the brain. Because of this, SLNs (which contain DPL) will be administered via the intranasal route [43].

The second concern is that in the previous section, they wrote that liposomes significantly reduce the cellular toxicity of donepezil. Does the donepezil systemic toxicity correlate with cellular toxicity.

Author Response: Dear Reviewer, thank you for remark. Liposomes significantly reduces the systemic toxicity. It is not a cellular toxicity.

Donepezil via oral drug delivery cannot cross BBB. Donepezil-loaded liposomes are made and intranasally administered to rapidly cross the BBB, showing improved bioavailability and reducing systemic toxicity [76].

Sentence in lines 171-172 (with ref. 44) is a repetition. The authors mentioned donepezil several times, and the manuscript is about the treatment of Alzheimer's disease, so this sentence (not the quote!) is unintelligible because of its obvious content.

 Author Response: Correction has been made

- Lines 177-178 contain a repetition in the essential oil - assumed - beneficial effects.

 Author Response: I agreed, now correction has been made

α-bisabolol is a sesquiterpene alcohol found in Matricaria chamomilla, it shows beneficial effects such as anti-cholinesterase, anti-plasmodial, anti-inflammatory, anti-cholinesterase, anti-cancer, and antimicrobial, but it has low solubility and bioavailability [46].

- Lines 185 and 186 contain two kinds of spelling. Please decide which one is correct and use that form in the text.

 Author Response:  Corrections have been made: It is Nanostructured Lipid Carriers (NLCs)

- In line 193, the authors again compare the sizes of something (smaller size) but have forgotten to inform the reader: comparing to what.

 Author Response: Thanks for remarks, now it has been corrected:

The NLCs have merits over conventional dosage because of their lipophilic nature and smaller size over the particle size of conventional dosage forms, which helps the drug molecules easily cross the BBB. The lipid matrix protects the NLCs from being degraded by enzymes and allows the active drug to reach the target site [52-53].

- Lines 206-209 contain a repetition. The reader will understand at first reading what is meant by the limited applicability of metal nanoparticles (="every metallic nanoparticle is not suitable for targeted drug delivery"), despite the grammatical challenges of the message in parentheses.

 Author Response: Corrections have been made

Metallic nanoparticles have been demonstrated as a useful therapeutic approach in managing AD via targeted drug delivery. It includes gold, silver, selenium, iron, and cerium and is known to have good anti-AD properties [29].

- As mentioned before, Section 3.4 has a better place after Section 3.2, as the polymeric nanoparticles are organic materials.

 Author Response: Now, polymeric nanoparticles have been placed after lipid nanoparticles

- Line 239 contains redundant information, as the whole manuscript is on a neurodegenerative disease.

 Author Response: Dear Reviewer, Thank you for your remark. It should be AD. Correction have been made.

- Message about the curcumin (lines 243-244) is also repeated there.

 Author Response: Text has been revised now

Curcumin is a phytoactives molecule. Researcher develops PLGA-coated curcumin NPs show an antioxidative effect and demolish Aβ aggregates in the AD animal model [59].

The authors confused the terms in the sentence about PLGA nanoparticles (lines 244-245). Not curcumin encapsulates PLGA, but vice versa. It is clear from the title of the cited article. Have the authors read that publication?

 Author Response: Dear Reviewer, thank you for worthy suggestions. Corrections have been made.

Researcher develops PLGA-coated curcumin NPs show an antioxidative effect and demolish Aβ aggregates in the AD animal model [59].

- In line 257, it is again unclear whether the FDA has approved the memantine API or the nanoformulation.

 Author Response: Correction have been made. Memantine is an FDA-approved drug used in the treatment of AD.

- In line 261, the authors discuss the emulsified chitosan/rivastigmine formulation. Why is it here and not in the next chapter, which discusses the emulsion-based nanocarriers?

 Author Response: Lipid polymer hybrid nanoparticles section contents are only polymeric nanoparticles. So, this has section has been merged with just above section (polymeric nanoparticles).

- In lines 287-288, it is unclear whether the AD-reducing property is a general character of the NSAIDs or the NSAIDs nanoformulation have this property only.

 Author Response: Dear Reviewer, thanks for your remarks. AD-reducing property observed with ibuprofen only and this supported with the reference [85].

Researchers found that people who took NSAIDs like ibuprofen had a lower risk of developing AD [85]. A novel repurposing strategy and route of administration are presented in this study for the treatment of AD. The in vivo result in rats found uptake of a novel ibuprofen-loaded ME nearly four times higher than that of the intravenous and 10 times that of the oral administrations [86].

- The name of compounds, except brand or specific cases, should not start with a capital letter inside a sentence (e.g., naringenin, NOT Naringenin).

 Author Response: Corrections have been made.

- The existence of Chapter 4 is questioned. The content of this section is effectively one of the concluding remarks and therefore fits better into the Conclusion. Generally, the title of the chapters in a manuscript is a recommendation and not a strict rule - except, of course, for a few necessary chapters, like the Introduction, Discussion, or Conclusion.

 Author Response: Dear Reviewer, thank you for your suggestion. Accordingly, placed in the conclusion section.

- In line 321, the administered sounds better than the given.

 Author Response: Correction have done

- Lines 346-351 are in the wrong location. These sentences are more appropriate in the Conclusions section, as they are not consistent with the title of the chapter.

 Author Response: Dear Reviewer, thank you for worthy suggestions. Now, it has been placed accordingly.

- As mentioned before, some journal abbreviations are missing (refs. 3, 7, 8, 9, 10, 12, 18, 19, 25, 26, 27, 30, 32, 33, 42, 43 - wrong punctuation mark -, 45, 51, 52, 55, 60, 62, 68, 70, 77, 81, 86). In case of doubts, please use Google or any other search engine. Please also note that if the journal name end is not an abbreviation, a dot is unnecessary between the name and the year. For example, refs 1 and 13 are correct, while ref 2 is not.

Author Response: Now, checked each reference. Corrections have been made at required places.

Comments on the Quality of English Language

Reformulation of some sentences is necessary, principally due to the redundant or repeated content. The manuscript contains many very long sentences which require modifications, too.

Author Response: Dear Reviewer, thank you for suggestions. Corrections have been made and modified the sentences.

Round 2

Reviewer 2 Report

The authors have significantly improved the quality of the manuscript. Unfortunately, one more round of revisions is necessary to correct the remaining minor flaws.

- The referee is afraid that mistakes are in lines 231-234. In the sentences (line 231), α-bisabolol should start with capital B. On the other hand, the expression "anti-cholinesterase" is twice in the sentence, as mentioned in the previous referee's report.

- The reference section still does not conform with the journal requirements.

The required format is "... Title. Journal Year, Volume, ...";

but the authors still use "... Title. Journal Volume (Year) ..."

The language quality seems adequate.

Author Response

 Reviewer 2

Comments and Suggestions for Authors

The authors have significantly improved the quality of the manuscript. Unfortunately, one more round of revisions is necessary to correct the remaining minor flaws.

 Author Response: Thank you for your remark.

- The referee is afraid that mistakes are in lines 231-234. In the sentences (line 231), α-bisabolol should start with capital B. On the other hand, the expression "anti-cholinesterase" is twice in the sentence, as mentioned in the previous referee's report.

Author Response: Corrections have been made. 

- The reference section still does not conform with the journal requirements.

The required format is "... Title. Journal Year, Volume, ...";

but the authors still use "... Title. Journal Volume (Year) ..."

Author Response: Dear Reviewer, Thank you for your remark. Corrections have been made